# High-Throughput Antibody Profiling Identifies Targets of Protective Immunity against *P. falciparum* Malaria in Thailand

**DOI:** 10.3390/biom13081267

**Published:** 2023-08-18

**Authors:** Ifra Hassan, Bernard N. Kanoi, Hikaru Nagaoka, Jetsumon Sattabongkot, Rachanee Udomsangpetch, Takafumi Tsuboi, Eizo Takashima

**Affiliations:** 1Division of Malaria Research, Proteo-Science Center, Ehime University, Matsuyama 790-8577, Japan; h870006u@mails.cc.ehime-u.ac.jp (I.H.); nhikvip@tmd.ac.jp (H.N.); 2Centre for Malaria Elimination, Institute of Tropical Medicine, Mount Kenya University, Thika 01000, Kenya; bkanoi@mku.ac.ke; 3Mahidol Vivax Research Unit, Faculty of Tropical Medicine, Mahidol University, Bangkok 10400, Thailand; 4Center for Research and Innovation, Faculty of Medical Technology, Mahidol University, Salaya, Nakhon Pathom 73170, Thailand; rachanee.udo@mahidol.edu; 5Division of Cell-Free Sciences, Proteo-Science Center, Ehime University, Matsuyama 790-8577, Japan; tsuboi.takafumi.mb@ehime-u.ac.jp

**Keywords:** malaria, wheat germ cell-free system, immunoscreening, AlphaScreen

## Abstract

Malaria poses a significant global health challenge, resulting in approximately 600,000 deaths each year. Individuals living in regions with endemic malaria have the potential to develop partial immunity, thanks in part to the presence of anti-plasmodium antibodies. As efforts are made to optimize and implement strategies to reduce malaria transmission and ultimately eliminate the disease, it is crucial to understand how these interventions impact naturally acquired protective immunity. To shed light on this, our study focused on assessing antibody responses to a carefully curated library of *P. falciparum* recombinant proteins (n = 691) using samples collected from individuals residing in a low-malaria-transmission region of Thailand. We conducted the antibody assays using the AlphaScreen system, a high-throughput homogeneous proximity-based bead assay that detects protein interactions. We observed that out of the 691 variable surface and merozoite stage proteins included in the library, antibodies to 268 antigens significantly correlated with the absence of symptomatic malaria in an univariate analysis. Notably, the most prominent antigens identified were *P. falciparum* erythrocyte membrane protein 1 (PfEMP1) domains. These results align with our previous research conducted in Uganda, suggesting that similar antigens like PfEMP1s might play a pivotal role in determining infection outcomes in diverse populations. To further our understanding, it remains critical to conduct functional characterization of these identified proteins, exploring their potential as correlates of protection or as targets for vaccine development.

## 1. Introduction

Malaria is a significant public health issue and a primary cause of illness in tropical and sub-tropical countries. There were approximately 247 million malaria cases in 2021, resulting in roughly 619,000 deaths [1]. Developing an effective malaria vaccine is a critical milestone in decreasing morbidity and mortality and supporting disease eradication efforts. This goal is supported by evidence demonstrating that individuals living in malaria-endemic regions develop partial antibody-mediated anti-disease immunity [2,3]. This immunity tends to increase with consecutive infections [4]. Since symptomatic malaria is primarily attributed to asexual parasite infections of *Plasmodium falciparum*, it is hypothesized that blood-stage antigens have the potential to stimulate strong immune responses that can be targeted for vaccine development [5]. This has driven the development of various methods for the discovery and development of new vaccine candidates.

Immuno-epidemiological investigations have been instrumental in detecting, recognizing, and characterizing numerous dominant vaccine candidates currently under development. In particular, immunoscreening techniques that permit the simultaneous assessment of various proteins as prospective vaccine candidates or immune correlates have emerged as an imperative strategy. Several proteins, such as merozoite surface protein (MSP) 1, MSP2, MSP3; apical membrane antigen 1 (AMA1) [6]; serine repeat antigen 5 (SERA5) [7]; and glutamate-rich protein (GLURP [8]), have been identified as vaccine candidates through such approaches. However, most of these candidates have had limited efficacy in clinical trials, primarily due to the extensive genetics polymorphism [9]. This has informed the need to discover other novel molecules that confer protective immunity for downstream characterization.

The production of *Plasmodium* proteins through heterologous expression systems has posed a significant challenge in identifying the proteins targeted by anti-malaria protective immunity. The wheat germ cell-free system (WGCFS) offers a eukaryotic alternative to expressing plasmodial proteins with broad applications in malaria studies [10]. In our early experiments, we generated soluble *P. falciparum* proteins and demonstrated that WGCFS-expressed proteins are greatly advantageous over conventional *Escherichia coli* system-expressed proteins in the pursuit of obtaining high yields of full-length proteins [11]. Furthermore, it was revealed that WGCFS-produced malaria proteins exhibit greater immunoreactivity to human immune sera compared to identical proteins produced through *E. coli* cell-free systems [12]. In addition, the AlphaScreen system, a homogeneous proximity-based bead assay that detects protein interactions, has been integrated to WGCFS and customized to offer high-throughput antibody assays.

Leveraging the availability of a well-annotated 3D7 *P. falciparum* genome, recent investigations have focused on identifying parasite proteins that are key targets of antibodies-mediated blood-stage immunity. This includes variable surface antigens (VSA) that are localized to the surface of infected red blood cells, such as *P. falciparum* erythrocyte membrane protein 1 (PfEMP1), repetitive interspersed family (RIFIN) proteins, sub-telomeric variable open reading frame (STEVOR) proteins, and surface-associated interspersed gene family (SURFIN) proteins, as well as other asexual-stage parasite proteins (here referred to as BSPs) [13]. In one such study, the evaluation of these proteins were conducted using serum samples collected from individuals living in a malaria-endemic region of Uganda, characterized by a high entomological inoculation rate (EIR) of more than 100 infective mosquito bites per year [13]. The analysis revealed that over 95% of the proteins examined exhibited immunoreactivity. Of the proteins tested, 22 induced antibody responses significantly associated with a reduced risk of malaria episodes, as determined by three different definitions of clinical malaria based on parasite density (1000/2500/5000 parasites/µL blood) along with the presence of fever [13]. Importantly, 20 of the 22 selected proteins belonged to VSAs.

Because of the global concerted efforts to reduce the burden of malaria, *P. falciparum* cases have significantly decreased in Southeast Asia and some African countries [14]. Thus, it is important to evaluate the effect of reduced malaria transmission on the *P. falciparum*-specific antibody responses in the affected populations. In line with this idea, we have previously profiled *P. falciparum* BSPs against clinical malaria by analyzing serum samples obtained from Thailand, a region of low malaria endemicity with an EIR of 1–3 infective bites/year [15,16]. In that study, we observed that antibody responses against MSP3, merozoite surface protein Duffy binding-like (MSPDBL) 1, reticulocyte binding protein 2 homolog b (RH2b), and MSP7 were significantly higher in asymptomatic than in symptomatic individuals [16]. However, the antigens used for the study did not include VSAs, which could be important seromarkers of reduced malaria episodes, as demonstrated elsewhere [13,17].

In this study, we sought to comprehensively assess the antimalarial immune profiles in residents of a low malaria transmission region in Thailand [16]. The analysis targeted 691 *P. falciparum* antigens, including VSAs and BSPs. We observed that most of the antigens were immunoreactive, and subsequently identified *P. falciparum* antigens associated with reduced risk of symptomatic malaria from a wider range of proteins.

## 2. Materials and Methods

### 2.1. Plasma Samples and Ethical Statements

From June 2001 to May 2005, we collected serum samples from residents of Kong Mong Tha, located in Kanchanaburi Province in western Thailand. These individuals were enrolled in an active follow-up study. An asymptomatic *Plasmodium falciparum* malaria (Asy) case was defined as having a detectable parasitic infection, following a microscopic examination of Giemsa-stained blood films, but with no fever (<37.5 °C) at the time of sample collection [15]. Although samples from 19 participants were collected, only 17 were available for the current assays due to serum volume limitation. The study received approval from the Ethics Committee of the Thai Ministry of Public Health and the Institutional Review Board of the Walter Reed Army Institute of Research (WRAIR 778; dated 14 April 2000) [15]. In addition, we collected serum samples from 21 adults diagnosed with uncomplicated symptomatic *P. falciparum* malaria (Sym) between May and June 2005. These individuals resided in Tak Province, adjacent to Kanchanaburi Province in western Thailand. All 21 Sym cases had at least one microscopically confirmed and recorded *P. falciparum* malaria episode before the one corresponding to the sample collection. Sym malaria was defined as the presence of fever (≥37.5 °C) during sample collection and any parasitemia detectable by microscopy. The Sym group study received ethical approval from the Faculty of Medicine’s ethical review committees at Ramathibodi Hospital, Mahidol University (ID: 09-46-10 dated 24 September 2003). All individuals who participated in this study provided written informed consent. The Institutional Review Boards of Ehime University Hospital, Japan, also approved the protocol for using the archived serum samples in Japan (Aidaiibyourin 1507005 dated 1 August 2015) [16].

### 2.2. Production of P. falciparum Biotinylated Protein Library Using Wheat Germ Cell-Free System (WGCFS)

The protein library used in this study included a total of 691 proteins, which represented various protein families of *P. falciparum*, such as blood stage proteins (BSP; n = 157), selected from our previous study based on their significant immunoreactivities [13], cysteine-rich inter-domain regions of *P. falciparum* erythrocyte membrane protein 1 (PfEMP1) (CIDR; n = 108), Duffy binding-like domains of PfEMP1 (DBL; n = 163), repetitive interspersed family (RIFIN) proteins (n = 178), sub-telomeric variable open reading frame (STEVOR) proteins (n = 53), and surface-associated interspersed gene family (SURFIN) proteins (n = 32). The target genes were cloned into the pEU plasmid vector (CellFree Sciences, Matsuyama, Japan) as previously described [10,17,18]. Mono-biotinylated recombinant proteins were synthesized in vitro with WGCFS using a semi-automated GenDecoder 1000 robotic protein synthesizer (CellFree Sciences). All protein sequences were derived from the *P. falciparum* 3D7 reference strain [13].

### 2.3. Quantitation of Protein-Specific Antibodies Using AlphaScreen

The AlphaScreen platform was used to quantify serum antibodies in malaria-exposed individuals using a previously validated protocol [10]. Briefly, the assay was automated using VIAFO 384 (Integra, Zizers, Switzerland) and performed in 384-well OptiPlate microtiter plates (PerkinElmer, Waltham, MA, USA). Non-purified biotinylated recombinant *P. falciparum* proteins (5 µL) were mixed with 4000-fold-diluted serum in reaction buffer (10 µL) and incubated for 30 min at 26 °C to form a protein–antibody complex. Suspension of streptavidin-coated donor-beads and acceptor-beads conjugated with Protein G in the reaction buffer was added to the mixture (10 µL), and the whole mixture was incubated in the dark for 1 h for optimal binding of the donor- and acceptor-beads to biotin and human antibody, respectively. The luminescence signal emitted by the protein–antibody complex at 620 nm was detected by the EnVision plate reader (PerkinElmer) and captured as raw AlphaScreen Counts (ASCraw). Biotin-SP-conjugated ChromPure human IgG (Jackson ImmunoResearch, West Grove, PA) was added to each plate to ensure day-to-day and plate-to-plate normalization. A 5-parameter logistic standardization curve was then used to generate normalized ASCnor, which is described as ASC in this paper. The biotinylated IgG was also used as the positive control, while translation mixtures with wheat germ extract (WGE) but lacking the parasite mRNA transcript were used as the negative control. To avoid any experimental bias, all assays were randomized [19].

### 2.4. Statistical Analysis

Data analyses were performed using R software (Version 4.0.1, R Foundation for Statistical Computing) and Prism 9 (GraphPad Software, LLC, Boston, MA, USA). Data obtained from the AlphaScreen were log-transformed before analysis. Statistical significance was defined as *p* < 0.05. Mann–Whitney test was performed to evaluate the statistical difference between median parasitemia of Asy vs. Sym groups. Spearman’s rank correlation test was used to evaluate the correlation between seroprevalences of Thai vs. Ugandan serum samples.

The protein seropositivity cut-off value to human sera was defined at the lowest non-negative ASC value from the assayed samples. A protein was considered immunoreactive if more than 10% of the volunteers had ASC levels above the seropositivity cut-off value [20]. To further explore the association between antibody responses and malaria outcomes, we performed logistic regressions with symptomatic and asymptomatic malaria outcomes as dependent variables. In multivariate regression, the analysis was adjusted for participant’s age and gender.

We used principal component analysis (PCA) to evaluate antibody responses to different proteins among individuals with or without clinical malaria. PCA essentially reduces the dimensionality of antibody responses by generating fewer composite variables to capture as much variance in that dataset. The model then assesses and compares the relationship between the scores obtained for each of these components per subject. We included all immunoreactive antigens to assess the overall variance among the antigens, regardless of their groupings. To identify PCA-derived clusters of antibody responses that were involved in protection against clinical malaria, we selected principal components where at least three variables were loaded and the eigenvalue was greater than 2 or the proportion of variance explained was > 5%. Individual contributions to the PCA were assessed in association with clinical malaria outcomes [13].

## 3. Results

### 3.1. Characteristics of the Malaria Participants

Serum samples were obtained from adult participants in Thailand, as shown in Table 1. The median age of the Asy group, 30 (range; 11–50) years, was higher than the Sym group, 23 (range: 17–50) years. Gender was biased toward males for Sym (90% male) than in the Asy group (58.82% male). Median parasitemia in the Asy group was significantly lower than in the Sym group: 0.003% (range 0.0006–0.2%) and 0.6% (range 0.0021–2.9%), respectively (Mann–Whitney U test, *p* < 0.0001).

### 3.2. Profiling of the Human IgG Responses to Plasmodium Falciparum Proteins

We profiled the human antibody responses to the library of malaria proteins using the Thai samples (n = 38). The geometric means of the AlphaScreen count (ASC) of all proteins were compared between Asy and Sym groups (Figure 1A). Of 691 proteins, 674 proteins, including cysteine-rich inter-domain regions (CIDR) (100%), surface-associated interspersed gene family (SURFIN) (100%), blood-stage proteins (BSP) (97.5%), Duffy binding-like domains (DBL) (96.9%), repetitive interspersed family (RIFIN) (97.2%), and sub-telomeric variable open reading frame (STEVOR) (94.3%), were immunoreactive, with a seroprevalence higher than the defined cut-off value of 10% (Figure 1B and Appendix A). The seroprevalence of the proteins in all assessed participants varied between the protein groups: BSP ranged from 0 to 100%, CIDR from 22 to 90%, DBL from 0 to 100%, RIFIN from 5 to 100%, STEVOR from 0 to 95%, and SURFIN from 26 to 100%; however, the difference was not significant. We did not observe a significant correlation between participants age with either the breadth of antibody responses (number of antigens recognized by an individual, Spearman rank correlation tests; r = 0.0157, *p* = 0.92) nor the geometric means of ASC of all antigens recognized by an individual (r = 0.1811, *p* = 0.27) (Appendix A).

### 3.3. Association between Antibody Responses and Risk of Clinical Malaria

To gain further insight into the relationship between the antibody levels and malaria outcomes, we performed univariate logistic regression analysis to determine the odds ratios (ORs) using antibody responses as the independent variable and clinical outcomes as the dependent variable. Of the 674 immunoreactive antigens, 268 were significantly associated with reduced risk of clinical malaria, with Ors less than 1, indicating a negative correlation between antibody responses and symptomatic malaria outcomes (Figure 2A, and ordered list of all proteins is included in Appendix A). The top 10 significant antigens selected by this univariate analysis mainly included PfEMP1s (DBL and CIDR; Figure 2B and Table 2). When we performed multivariate analysis, with antibody responses as the independent variable and symptomatic outcomes as the dependent variable while adjusting for age and gender, the association between antibody responses and clinical outcomes was diminished. However, 247 antigens remained significant, with Ors of less than 1. Again, the top significant antigens selected by multivariate analysis were similar to those selected by univariate analysis (Appendix A, Appendix A), indicating the important role of VSAs in malaria-acquired immunity. The ORs of two RIFINs (PF3D7_1400400; RS215 and PF3D7_0732400; RS103) were significantly greater than 1 (Appendix A), indicating that these proteins may be acting differently when compared to the other proteins.

### 3.4. Principal Component Analysis of Antibody Responses

Since the antibody responses and their corresponding protective immunity involve numerous antibodies acting simultaneously but targeting different proteins, we performed principal component analysis (PCA) with different proteins to capture the effect of all antibodies in a single analysis and extract important associations [13]. The analysis indicated that the first three components (with eigenvalues of 238, 106, and 34, respectively) accounted for 54.4% of the total variation in the data (Figure 3A). Dim 1 vs. 2 (Figure 3B) explained 49.5% of the variability, and the main contributors were PF3D7_0808700 (domain CIDRα3.1), PF3D7_0632500_DBLγ1, PF3D7_1100200_DBLβ9, PF3D7_1035600, PF3D7_0700100_CIDRα3.1. Dim 1 vs. 3 (Figure 3C) explained 39.1% of data variability, with the strongest loading arising from antibodies against PF3D7_0808700_CIDRα3.1, PF3D7_0632500_DBLγ1, PF3D7_1100200_DBLβ9, PF3D7_0413100_DBLδ1, PF3D7_0733000_CIDRβ1. Dim 2 vs. 3 (Figure 3D) was mainly reflective of PF3D7_1477600, PF3D7_1040600, PF3D7_1040400, PF3D7_0500500, and PF3D7_0532900 and accounted for 20.2% of the variation.

### 3.5. Comparison of Seropositivity between Thai and Ugandan Samples

We sought to further investigate how to leverage samples from multiple regions to identify *P. falciparum* antigens associated with reduced risk of symptomatic malaria. To do this, we compared the antibody levels in blood samples from Thailand—this study—and from our recent study in Uganda [13]. In both studies, an antigen was defined as immunoreactive if present in more than 10% of samples assayed [13]. Of all the antigens measured in both studies (579 of 691), 566 antigens were commonly found to be immunoreactive. There was a significant positive correlation between the seroprevalence percentages obtained from samples collected in Thailand and Uganda (Figure 4A). Among the different groups of antigens, the seroprevalences against blood-stage proteins (BSP) exhibited the highest level of correlation. (Figure 4B). Unexpectedly, there was a significant correlation observed among the seroprevalences against *P. falciparum* erythrocyte membrane protein 1 (PfEMP1) domains as well (Figure 4C,D). In particular, a strong correlation was found between the seroprevalence percentages of the overlapping immunoreactive antigens (n = 163) originating from the DBL domain in both sample sets (Figure 4B, r = 0.7219, *p* < 0.0001).

## 4. Discussion

Residents of malaria-endemic regions develop immunity to life-threatening malaria complications after repeated exposure [21]. To identify target molecules of the acquired immunity, protein microarrays with printed proteins or peptides synthesized using an *Escherichia coli* cell-free system have been widely utilized. This approach could identify seroreactivity to some *Plasmodium falciparum* erythrocyte membrane protein 1 (PfEMP1) domains [22], blood-stage proteins (BSPs) [23], or proteome-wide proteins [24]. In addition, KILchip v1.0, a protein microarray printed with 109 or 111 purified *P. falciparum* antigens expressed mostly by mammalian Expi293 cells has been reported [25]. These assays revealed that the antibody kinetics upon natural *P. falciparum* infection are useful for the identification of recent infections [26]. The major challenge with the *E. coli* cell-free system is that the synthesized proteins exhibit low immunoreactivities with sera derived from malaria-exposed individuals, suggesting incorrect folding of the proteins [23,27]. On the other hand, it is difficult to scale up the numbers of antigens prepared using the mammalian cells. The wheat germ cell-free system (WGCFS) has been used to synthesize large numbers of proteins, including plasmodial proteins [28]. It was reported that malaria proteins synthesized using the WGCFS have higher immunoreactivity to human immune sera as opposed to identical proteins synthesized with the *E. coli* cell-free system [12].

AlphaScreen has been widely used for the detection of protein–protein, protein–DNA, and protein–RNA interactions and for the screening of proteins interacting with the target proteins, inhibitors of the target complex of proteins, and so on [29]. Because AlphaScreen has been used for the detection of protein–protein interactions, including antigen–antibody interactions, with robust results observed especially when used in combination with the biotinylated protein library produced by WGCFS [30,31,32], we applied this method in our studies [10].

Due to the collective global efforts to alleviate the impact of malaria, the prevalence of *P. falciparum* infections has considerably decreased in several African countries and Southeast Asia [14]. As a result, it is crucial to assess the impact of reduced malaria transmission on the *P. falciparum*-specific antibody responses in the affected populations. Thus, this study aimed to comprehensively identify *P. falciparum* antigens that are the targets of naturally acquired immunity with serum samples taken from Thailand, a region of low malaria endemicity. Antibody responses to 268 out of 691 antigens were positively associated with the reduced risk of symptomatic malaria. Most antigens selected were PfEMP1 domains (Table 2, Appendix A, Figure 3), suggesting that the PfEMP1s are a major target of naturally acquired antibodies associated with reduced risk of clinical malaria. These observations were in good agreement with our previous findings in Ugandan samples from participants aged 6–20 years [13], suggesting that even in low-malaria-transmission areas, individuals acquire antibodies to similar antigens as those living in high-transmission areas. The top 10 antigens that were significantly associated with reduced risk of clinical malaria in the univariate analysis included mainly PfEMP1 domains, namely Duffy binding-like (DBL) and cysteine-rich inter-domain region (CIDR) (Figure 2, Table 2 and Appendix A). In contrast, there was only one BSP, Merozoite Surface Protein 6 (MSP6), selected among the top 10 antigens. These data suggested that the PfEMP1s have higher immunoreactivity than BSPs, even in the low-endemic regions. However, the small sample size used in this study may have under- or over-estimated immune associations; thus, further studies with larger sample sizes or a targeted study with fewer proteins are required to confirm the current results.

We found a significant positive correlation of seroprevalence for *P. falciparum* antigens recognized by Thai and Ugandan serum samples (Figure 4A). Moreover, a stronger correlation was found with PfEMP1 domains (Figure 4C), even though they are very polymorphic antigens. Since the sequences from the *P. falciparum* 3D7 strain were used to produce the recombinant proteins in this study, these results could suggest that the immunoreactivities of PfEMP1s were derived from the cross-reactivities among the domains. The strong correlation of seroprevalence between Thai and Ugandan serum samples indicated that the reactivities are mainly determined by the characteristics of the target antigen, i.e., conservation, repetitiveness, expression levels, and/or copy numbers of similar domains, rather than the host background. In addition, a recent study indicated that there are many “inter-protein motifs” of cross-reactive peptide sequences in PfEMP1s [33]. Although cross-reactivity may interfere with developing high-affinity specific antibodies [34], it could also indicate cross-protection. Similarly, in vivax malaria, a strong correlation in seroprevalence against 300 *P. vivax* antigens among Thai, Brazilian, and Papua New Guinean (PNG) samples was also observed [20], indicating that the antibody reactivities are mainly determined by the antigen but not by the host background.

The identification of a domain of PfEMP1, DBLβ3_D4 of PF3D7_1150400, as one of the top-ranked proteins in this study was very encouraging for several reasons. First, the results were consistent with a study showing that peptides and recombinant proteins from the domain reacted higher in asymptomatic malaria than severe malaria samples obtained from Benin [35]. Second, the domain was also identified in studies in Uganda, where the domain-specific antibody levels were associated with protection against clinical malaria [16]. Although the sites in these studies are located on different regions with different malaria endemicity and host genetic backgrounds [36], the data could indicate that the same PfEMP1 domain is an important factor in immunity against clinical malaria. Functionally, the PF3D7_1150400 DBLβ3_D4 domain is known to bind to the intercellular adhesion molecule 1 (ICAM-1) receptor on the host endothelial cell and could be involved in parasite sequestration [37]. Indeed, antibodies targeting DBLβ3 have been shown to inhibit PfEMP1 binding to ICAM-1 and exhibit cross-reactivity to domain cassette 4 (DC4) from genetically distinct parasite isolates, indicating a broad inhibitory effect. The significance of anti-DBLβ3 antibodies in protective immunity was also highlighted in previous analyses [24]. Additionally, PF3D7_1150400 protein also has a domain cassette DC13, which is composed of the tandem of domains DBLα1.7 and CIDRα1.4 and binds to Endothelial protein C receptor (EPCR) receptor through CIDRα1.4 [38]. Research has demonstrated that the presence of the EPCR and ICAM-1 binding activities on a single protein plays a critical role in the pathogenesis of cerebral malaria [39]. Taken together, these data suggest that even in regions of low malaria transmission settings, one can also obtain semi-immunity after only a few infections [3]. Further characterizations of PfEMP1 molecules with these domains would be of interest to contribute to developing a PfEMP1-based blood-stage malaria vaccine.

PCA is a valuable tool for analyzing complex datasets involving multi-dimensional data, mixed infections, and complex host-parasite interactions. Using PCA, we can explore the joint effects of multiple variables and identify underlying patterns that may not be obvious [13]. Here, the approach allowed us to capture the combined effects of all antibodies in a single analysis, which is particularly relevant given that protective antibody responses are likely to involve many antibodies working together. With the current dataset, the first three principal components accounted for the majority of the outcomes observed. The main contributors for Dim 1 vs. 2, and Dim 1 vs. 3 seem to contribute to the protective immunity because the vectors head for the cluster formed by Asy cases. Basically, antibodies against PfEMP1 domains were the major contributing variables (Figure 3), which is consistent with the results of logistic regression analysis. Out of them, PfEMP1s (PF3D7_0632500_DBLγ1, and PF3D7_0413100_DBLδ1) were identified as the main contributors to the protective immunity and were also the case for Ugandan samples [13]. In contrast, those of Dim 2 vs. 3, mainly antibodies against repetitive interspersed family (RIFIN) proteins, head for the cluster formed by Sym cases, suggesting that the antibodies would not contribute to the immunity or instead promote the symptoms. This was also consistent with the findings that antibody levels against 2 RIFINs are significantly positively correlated with occurrence of symptomatic malaria by logistic regression. In contrast, our previous study indicated that antibodies against selected RIFINs contribute to protective immunity [17]. This could not be explained by the fact that RIFINs bind to leucocyte immunoglobulin-like receptor B1 (LILRB1)-containing antibodies, which are important mechanism for immune evasion [40,41]. Since the function of most of the selected RIFINs remains unknown, further studies are needed for elucidating the function of anti-RIFINs antibodies.

## 5. Conclusions

In summary, in regions of differing malaria transmission, and differing human and parasite genetic backgrounds, host immune responses targeting specific parasite antigens could play a crucial role in determining the outcomes of *P. falciparum* infections. The antigens identified by this study could serve as valuable serological markers for identifying asymptomatic infections and as potential candidates for blood-stage vaccines. Thus, to gain detailed knowledge of protective immunity against symptomatic malaria, functional characterization of the selected proteins, and the antibodies against them, needs to be elucidated.

## Figures and Tables

**Figure 1 biomolecules-13-01267-f001:**
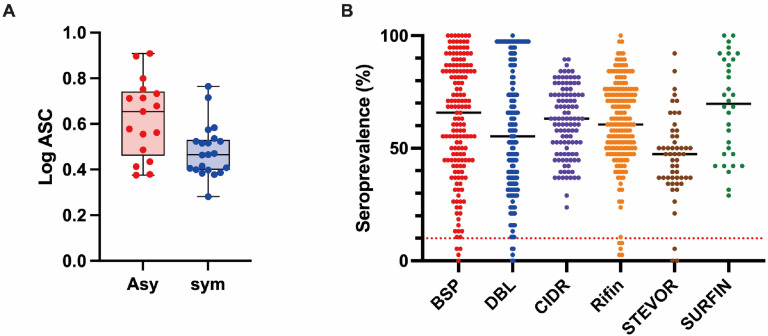
Immunoreactivity of the *P. falciparum* antigens against Thai serum samples. (**A**) The median AlphaScreen counts (ASC), indicating antibody responses against *P. falciparum* antigens, were compared between two sample groups: Asymptomatic (Asy, n = 17) and Symptomatic (Sym, n = 21) groups. The median reactivity of individuals is represented by dots, while the median of each group is depicted by black bars. (**B**) The seroprevalence of the specified *P. falciparum* proteins was determined using Thai serum samples (n = 38). The median seroprevalence of the group is represented by black bars. A dashed red horizontal line at 10% seroprevalence serves as the cutoff value, indicating the threshold for defining an immunoreactive antigen used for subsequent statistical analysis. Out of the 691 proteins tested, 674 exhibited a seroprevalence above 10% seroprevalence.

**Figure 2 biomolecules-13-01267-f002:**
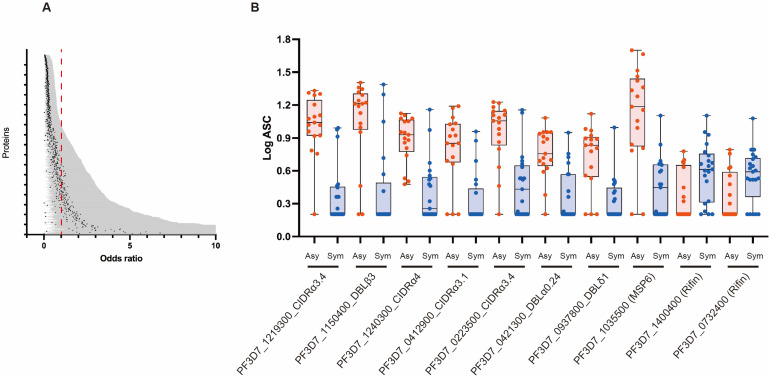
Comparison of the antibody responses with the risk of clinical malaria. (**A**) The association between antibody responses and the risk of clinical malaria as analyzed by univariate logistic regression. Black dots indicate the odd ratios, and error bars indicate a 95% confidence interval, and the red vertical dashed line represents an odds ratio of 1. The 268 antigens exhibited a significant negative association with malaria outcomes shown in the upper section. The very high upper confidence intervals derived from 37 antigens at the bottom are truncated in the figure. The complete list of all proteins is included in Appendix A. (**B**) The antibody levels between the Asymptomatic (Asy) and Symptomatic (Sym) groups were compared. The log 10-transformed AlphaScreen Count (ASC) for the 10 selected antigens are represented by dots and box plots, namely PF3D7_1219300_CIDRα3.4; DC239, PF3D7_1150400_DBLβ3; DC217, PF3D7_1240300_CIDRα4; DC243, PF3D7_0412900_CIDRα3.1; DC47, PF3D7_0223500_CIDRα3.4; DC15, PF3D7_0421300_DBLα0.24; DC66, PF3D7_0937800_DBLδ1; DC195, PF3D7_1035500 (MSP6); WE18, PF3D7_1400400; RS215 and PF3D7_0732400; RS103. Blue and red indicate Asy and Sym cases, respectively. The antigens’ names consist of PlasmoDB gene ID plus the expressed domain or protein name.

**Figure 3 biomolecules-13-01267-f003:**
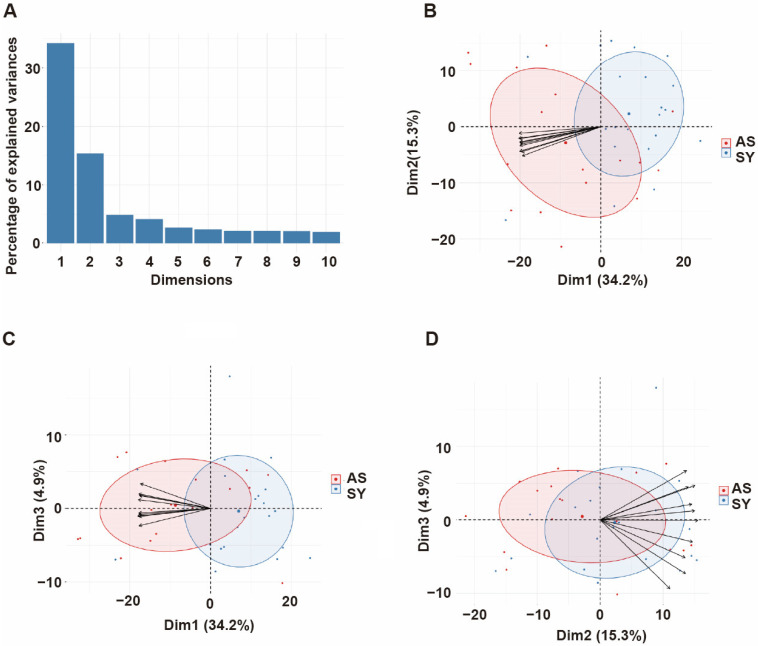
Principal components analysis (PCA) of antibody responses. (**A**) Distribution of principal components (first 10 dimensions) that account for the highest variance in antibody responses derived from all immunoreactive proteins. (**B**) Plots displaying the distribution of principal components (PCs) based on their antibody responses. The PCs are illustrated by Dim 1 vs. 2, (**C**) Dim 1 vs. 3, and (**D**) Dim 2 vs. 3, which explain the antibody responses (percentage in parentheses). Red and blue dots represent Asy, and Sym cases, respectively. Light red and blue ellipses indicate the distribution of Asy and Sym cases, respectively. Black arrows indicate the direction of maximum increase and strength (through the length) of the first 10 antigens that contributed to the principal components to the overall distribution.

**Figure 4 biomolecules-13-01267-f004:**
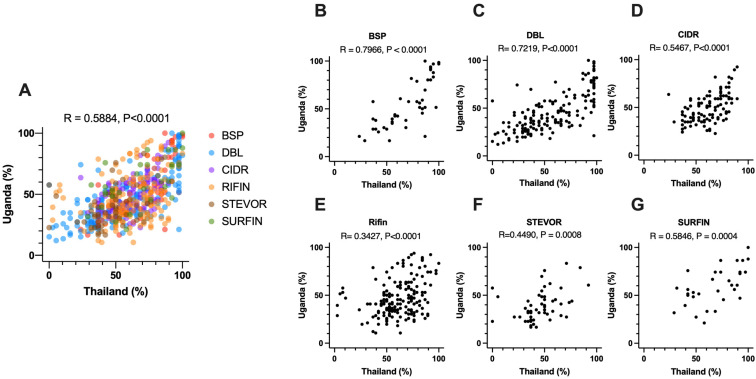
The seroprevalence % for antigens recognized by Thai and Ugandan serum samples were compared via Spearman rank correlation test. The comparison of the seroprevalence for (**A**) all antigens, the groups of antigens are shown by the indicated colors, (**B**) BSP, (**C**) DBL, (**D**) CIDR, (**E**) RIFIN, (**F**) STEVOR, (**G**) SURFIN.

**Table 1 biomolecules-13-01267-t001:** Characteristics of participants enrolled in the study in Thailand.

Characteristics	Asy (n = 17)	Sym (n = 21)
**Age in years at sampling, median (range)**	30 (11–50)	23 (17–50)
**Gender (male %)**	58	90
**Median parasitemia (range) %**	0.003 (0.0006–0.2)	0.6 (0.0021–2.9)

**Table 2 biomolecules-13-01267-t002:** Association between antibody responses and risk of clinical malaria (univariate analysis).

	Gene ID and Domain	Protein ID	OR	Lower CI	Upper CI	*p* Value
1	PF3D7_1219300_CIDRα3.4	DC239	0.002882631	0.000123653	0.067200484	0.000272011
2	PF3D7_1150400_DBLβ3	DC217	0.037865179	0.006190304	0.231615732	0.000395639
3	PF3D7_1240300_CIDRα4	DC243	0.002350607	0.0000779	0.070905936	0.000496605
4	PF3D7_0412900_CIDRα3.1	DC47	0.009541722	0.000692251	0.131519501	0.000509778
5	PF3D7_0223500_CIDRα3.4	DC15	0.008724198	0.000563243	0.135131077	0.000694702
6	PF3D7_0421300_DBLα0.24	DC66	0.002910712	0.0000985	0.085977897	0.000723691
7	PF3D7_0937800_DBLδ1	DC195	0.004367111	0.000178979	0.106558101	0.000856835
8	PF3D7_1035500 (MSP6)	WE18	0.011459908	0.000822541	0.159663158	0.000883835
9	PF3D7_0425800_CIDRβ5	DC76	0.016955413	0.001510087	0.19037717	0.000952049
10	PF3D7_1366400 (RHOP148)	WE76.2	0.006899884	0.00035859	0.132765465	0.000972586

## Data Availability

All the data used in this study and R-scripts used for the statistical analysis are available from the authors on request.

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
