# Peer review of "High-Throughput Antibody Profiling Identifies Targets of Protective Immunity against *P. falciparum* Malaria in Thailand"

_biomolecules, 2023, doi:10.3390/biom13081267_

Round 1

Reviewer 1 Report

Congratulations to the authors for the article that aimed ti identify the P.falciparum antigens targets of naturally acquired immunity with serum samples from individuals in Thailand, a region of low malaria endemicity. 

The methodology is well described and responds to the proposed objective, however I observe that some non-parametric  tests used in results not indicated in the methodology.  The Man-Whithey (U) test indicated in lines 165 to 167 to compare statistical difference  between the Asy and Sym groups. Spearma's non parametric correlation test to check form seroprevalence correlation for antigens recognized by serum samples from Thailand and Uganda. 

Still in results, in lines 199 and 200, and also 219 and 220 refer to their discussion. In these lines the authors make suggestions. In the paragraph reffering to lines 199 and 200 the authors even use the term nevertheless, line 195. 

As for the discussion, it is very well done and in fact discussing the results presented here. 

These are the observations I make for the moment.

Author Response

Congratulations to the authors for the article that aimed ti identify the P.falciparum antigens targets of naturally acquired immunity with serum samples from individuals in Thailand, a region of low malaria endemicity. 

The methodology is well described and responds to the proposed objective, however I observe that some non-parametric  tests used in results not indicated in the methodology.  The Man-Whithey (U) test indicated in lines 165 to 167 to compare statistical difference  between the Asy and Sym groups. Spearma's non parametric correlation test to check form seroprevalence correlation for antigens recognized by serum samples from Thailand and Uganda. 

Thank you for pointing this out, It’s now included in the updated manuscript.

Still in results, in lines 199 and 200, and also 219 and 220 refer to their discussion. In these lines the authors make suggestions. In the paragraph reffering to lines 199 and 200 the authors even use the term nevertheless, line 195. 

We have carefully edited the lines (199, 200, 219, 220 and 195) for clarity.

As for the discussion, it is very well done and in fact discussing the results presented here. 

These are the observations I make for the moment.

Thank you for appreciating our work.

Reviewer 2 Report

This manuscript presents the results of extensive research conducted by the authors in the field of studying the immune profiles related to the pressing issue of antimalarial treatment.

While reviewing this work, I encountered the following comments:

Section "Plasma Samples and Ethical Statements": I kindly request more detailed information regarding the ethics clearance for both sample cohorts, including the date of issue and protocol number.

Since all samples were collected no later than 2005, is there a risk that their long-term storage may have led to a loss of their original immune profile? Consequently, are the authors studying a shifted pattern compared to the original one?

Table 1, bottom right cell: Can the median truly be 0.6 for a range that extends from 0.0091 to 0.5?

Section 3.2, figures, and tables: I would recommend integrating the figures and tables into the main body of the text rather than placing them in a separate section. Presenting each figure after its first mention in the text would enhance clarity.

I suggest dedicating a separate comprehensive section to "Conclusions." This section should explicitly outline the key findings of this study and provide more detailed insights into future perspectives.

Considering the valuable dataset obtained in this study, I strongly urge the authors to make the original data openly available, enabling other researchers to analyze them freely. This can be achieved by including the data in the Supplementary Materials or by employing any other appropriate means.

Author Response

This manuscript presents the results of extensive research conducted by the authors in the field of studying the immune profiles related to the pressing issue of antimalarial treatment.

While reviewing this work, I encountered the following comments:

Section "Plasma Samples and Ethical Statements": I kindly request more detailed information regarding the ethics clearance for both sample cohorts, including the date of issue and protocol number.

The approval details are now included in the updated text.

“The study received approval from the Ethics Committee of the Thai Ministry of Public Health and the Institutional Review Board of the Walter Reed Army Institute of Research (WRAIR 778; dated 14 April 2000) [15].”

And

“The study received ethical approval from the Faculty of Medicine's ethical review committees at Ramathibodi Hospital, Mahidol University (ID: 09-46-10 dated 24 September 2003), for the Sym group. All individuals who participated in this study provided written informed consent. The Institutional Review Boards of Ehime University Hospital, Japan, also approved the protocol for using the archived serum samples (Aidaiibyourin 1507005 dated 1 August 2015) [16].”

Since all samples were collected no later than 2005, is there a risk that their long-term storage may have led to a loss of their original immune profile? Consequently, are the authors studying a shifted pattern compared to the original one?

In order to protect the integrity of the samples, the samples were collected and shipped to Ehime University Japan, where they were kept as single-use aliquots at -80˚C that were only thawed for the current assay.

Table 1, bottom right cell: Can the median truly be 0.6 for a range that extends from 0.0091 to 0.5?

Thank you very much for pointing this out. Now the ranges were updated to the correct values.

Section 3.2, figures, and tables: I would recommend integrating the figures and tables into the main body of the text rather than placing them in a separate section. Presenting each figure after its first mention in the text would enhance clarity.

Thank you for your suggestion. It’s now updated in the revised manuscript.

I suggest dedicating a separate comprehensive section to "Conclusions." This section should explicitly outline the key findings of this study and provide more detailed insights into future perspectives.

This section has been edited as a subheading.  

Considering the valuable dataset obtained in this study, I strongly urge the authors to make the original data openly available, enabling other researchers to analyze them freely. This can be achieved by including the data in the Supplementary Materials or by employing any other appropriate means.

In the manuscript, we have added a statement that data is available from the authors on request.

Reviewer 3 Report

Manuscript titled ” High-throughput antibody profiling identifies targets of protective immunity against P. falciparum malaria in Thailand” aimed at assessing the antibody response in individuals from a low transmission area for malaria in Thailand.

With the use of their pre-validated library of P. falciparum recombinant proteins (n=691), authors were able to investigate which peptides / proteins were mostly immunogenic by asymptomatic  and symptomatic malaria patients. Authors had great intent but there are major issued that need to be address so that we can value the research presented in the manuscript. Overall the manuscript is often difficult to read as the language is difficult, awkward, vague or details are really missing. Often there are areas that make the manuscript difficult to follow as it is very convoluted in sentence structure and themes. Mostly the results are not descriptive enough to have the impact or clarity that is needed to justify many statements made from the authors.

More details are in the following:

Difficult language or other issues in Abstract: Please reword

*As the consented efforts optimized towards decreasing malaria transmission and for eventual malaria elimination are implemented, driving previously malaria high to low-transmission regions, it is necessary to understand how interventions affect the naturally acquired protective immunity.

*samples collected from residents of a malaria low-transmission region of Thailand

*We observed that of the variable surface antigens and merozoites stage proteins included in the library,

antibodies to 268 antigens significantly correlated to the absence of symptomatic malaria in univari-

ate analysis.

*Why mention the following? “The antibody assays were conducted using the AlphaScreen system” If important enough to be put into the abstract than authors need to describe what this system is.

* The top screened antigens were mainly PfEMP1 domains. What do you mean by screened here as it probable does mean what you want. Do you mean the antigens with the most immune responses against it mainly PfEMO1 domains. Or?

Please define what PfEMO1 domains are.

Too vague statement. Need to add context or explicit meaning “These observations were consistent with our recent findings with samples from a malaria hyper-endemic region, suggesting that similar antigens could be central in determining infection outcomes in diverse populations.”

For instance, what is the “recent findings” exactly?  Which similar antigens?

* It remains important to functionally characterize the unknown proteins as potential correlates of protection or as vaccine targets. What do you mean by unknown proteins

Introduction

Clarity is needed with explicit context and better language use:

*Developing an effective malaria vaccine is a crucial objective… Crucial objective for whom?

*This goal is supported by evidence demonstrating that individuals living in regions where malaria is endemic develop a partial immunity mediated by antibodies [2, 3]. Goal to develop a vaccine or what do the authors mean here as this statement does not make sense-

*Awkward sentence that does not make sense. Since symptomatic manifestation of malaria is primarily attributed to asexual blood-stage antigens of Plasmodium falciparum, which have the potential to stimulate a strong immune reaction, these discoveries have led to the implementation of various methods for exploring discovery and development of new vaccine candidates.

*Citations are needed in these sentences: Targeted immuno-epidemiological investigations have been instrumental in detecting, recognizing, and characterizing numerous dominant vaccine candidates currently 45

under development. In particular, immunoscreening techniques that permit the simulta- 46

neous assessment of various proteins as prospective vaccine candidates or immune corre- 47

lates have emerged as a crucial strategy.

*Grammar in this sentence and more explicit understanding of what these proteins are, is needed, for example spell out the acronymsSeveral proteins, such as MSP1, MSP2, MSP3, AMA1 Reviewed in [6], SERA5 [7] and GLURP [8] have been identified through such approaches.

*More description is needed for the AlphaScreen platform Line 54.

*Citation is needed in the following:

Recent investigations have focused on identifying parasite protein families that are localized to the surface 56

of infected red blood cells, i.e., variable surface antigens (VSAs), including all the domains 57

of erythrocyte membrane protein 1 (PfEMP1), repetitive interspersed family (RIFINs) pro- 58

teins, sub telomeric variable open reading frame (STEVORs), and surface-associated in- 59

terspersed gene family (SURFINs), as well as blood stage proteins (BSP) encoded in the 60

3D7 strain genome.

*Confused as to whom did the following? Was it the authors or others, orIf others, citation is needed and need to not use short hand  (and in other areas of the paper) here as it is more difficult to read. Reword please.

These proteins were evaluated against serum samples obtained from individuals residing in a malaria-endemic region of Uganda, where an entomological inoculation rate (EIR) of > 100 infective bites/year.

* This sentence is confusing. What is meant by “three different clinical malaria definitions?”

Of the proteins tested, 22 induced antibody responses significantly linked to a reduced risk of malaria episodes, as determined by three different clinical malaria definitions based on parasite density (1,000/2,500/5,000 parasites/μl blood) along with the presence of fever [11].

* Why is it important to evaluate…. In the sentence below?

It’s important to evaluate the effect of reduced malaria transmission on the P. falciparum-specific antibody responses in the affected populations.

*Punctuation is wrong in the sentence. Please correct. And citation [14] should be added to the data in this sentence.

In line with this idea, we have previously profiled P. falciparum BSPs against clinical malaria by analyzing serum samples obtained from Thailand; a region of low malaria endemicity where EIR of 1–3 infective bites/year [13].

*Confusing statements.  Why are the authors looking into VSA when they have cited their own work in the sentence and already looking into this?  Please really explain what is the point of looking into VSA if you have already done this as cited by yourselves.

However, the antigens used for the study did not include VSAs, which could be promising seromarkers of reduced risk of malaria episodes proved by our recent studies [11, 15].

And

we sought to comprehensively assess the antimalarial immune profiles in residents of a low malaria transmission region in Thailand as used previously [14].

 And

The analysis targeted 691 P. falciparum antigens, including VSAs and BSPs, with the AlphaScreen platform.

*What are the 691 antigens exactly?

*Why use AlphaScreen platform and not other systems?

Methods

Methods sections needs to be written so some can repeat them and reproduce the results demonstrated from the authors. Right now this is not the case. Controls are missing, explicit details are missing in most methods and information on repeats are not provided. Also authors must clarify if proteins were used or peptides in all parts of the manuscript.

More details in the following:

Awkward sentences. Please clarify.

*Do you mean “The absence of fever was used to characterize ……

The absence of fever characterized asymptomatic P. falciparum malaria (< 37.5 ℃) during the time of sample collection (Asy) [13].

* Although, in our previous study [14], we used 19 serum samples in Asy group, the number

of serum samples with enough amount left is 17 for this study.

* before the one corresponding to the sample collection.

* What is the ethical approval number. Please put like others here in the methods section.

The Institutional Review Boards of Ehime University Hospital, Japan, 104

also approved the protocol for using the archived serum samples [14].

*What is (9) in this statement?  Is it the citation, if so then put it correctly in the text.

The protein library used in this study included a total of 691 proteins, which repre- 108

sented various protein families of P. falciparum, such as blood stage proteins (BSP; n=157) 109

selected from our previous study based on their significant immunoreactivities (9), cyste- 110

ine-rich inter-domain regions of P. falciparum erythrocyte membrane protein 1 111

(PfEMP1)(CIDR; n=108), Duffy binding-like domains of PfEMP1 (DBL; n=163), repetitive 112

interspersed family proteins (RIFIN; n=178), subtelomeric variable open reading frame 113

proteins (STEVOR; n=53) and surface-associated interspersed gene family proteins 114

(SURFIN; n=32).

*Authors stated

The analysis targeted 691 P. falciparum antigens, including VSAs and BSPs, with the AlphaS- 81

creen platform.” But in the methods sections, the VSAs are not defined. Why?

*What genes were cloned and how? PCR protocol, Primers used, restriction enzyme used etc…

The target genes were cloned into the pEU plasmid vector (CellFree Sciences, Matsuyama, Japan).

*Italicize in vitro.

* How and which ones because why clone things if you are synthetically producing the proteins?  Or provide a citation.

Mono-biotinylated recombinant proteins were synthesized in vitro using a semi-automated GenDecoder 1000 robotic protein synthesizer (CellFree Sciences).

*Is there a citation for the AlphaScreen protocol?  If not then why the 1/4000 dilution of sera with proteins?  What concentration of the proteins were used? What are the positive controls for the binding experiment and negative controls for the experiment? What was the buffers used etc. As for the secondary antibody, what was its dilution, its controls that it is working and buffer used.

*Why was the following used as the cut-off mark as this is not the convention?

The protein seropositivity cut-off value to human sera was defined at the lowest 141

non-negative ASC value from the assayed samples.

*and why did the authors have this evaluation?

A protein was considered immunore-active if more than 10% of the volunteers had ASC levels above the seropositivity cut-off 143

Value.

*Where are the healthy control group in this study?

Results

In general results should be easy to follow by having the figure, table or supplementary data cited in the text and this is not the case in most places in the result section. Authors have just put at the end of the results the figures and tables and this makes it very difficult to follow the authors.

In addition, the legend should be descriptive enough that the figure or table can stand on its own and be understood. Please improve the legends. In addition, Figures and tables must be place near the text when it is first mention. Reader should not have to go down three pages to see the evidence in the result text. Lastly, it is not enough to superficially mention the result as in the following “The top significant antigens selected in the univariate analysis included PfEMP1s (DBL and CIDR) (Figure 2, Table 2, Table S2).

Authors need to describe the results so that we can understand their meaning and significance. Tables and figures should also be descriptive for this too. For example, why do the authors state this that the top….. include PfEMP….? Be exact as figure 2 has two panels and B) only mentions ten proteins where as Table 2 has 15 peptides. Why are there not 15 proteins mentioned in figure 2? Same feedback for all the results.

But Authors did well in describing the following, and every panel was describes:

The analysis indicated 205

that the first three components (with eigenvalues of 238, 106, and 34, respectively) ac- 206

counted for 54.4% of the total variation in this data (Figure 3A). Dim 1 vs 2 (Figure 3B) 207

explained 49.5 % of the variability and the main contributors were 208

PF3D7_0632500_DBLγ1 (ID#DC109), PF3D7_1240300_CIDRβ1 (ID#DC246), 209

PF3D7_0712900_CIDRβ1 (ID#DC149), PF3D7_0800300_CIDRα1.6 (ID#DC166), 210

PF3D7_0712600_CIDRα3.1 (ID#DC139). Dim 1 vs 3 (Figure 3C) explained 39.1% of data 211

variability, with the strongest loading arising from antibodies against 212

PF3D7_0632500_DBLγ1 (ID#DC109), PF3D7_1240300_CIDRβ1 (ID#DC246), 213

PF3D7_0425800_CIDRα1.6 (ID#DC71), PF3D7_0712600_CIDRα3.1 (ID#DC139), 214

PF3D7_1300300_CIDRβ3 (ID#DC268). Dim 2 vs 3 (Figure 3D) was mainly reflective of 215

PF3D7_0402100 (ID#EH9, Plasmodium exported protein PHISTb, unknown function), 216

PF3D7_1040400 (ID#RS156, Rifin), PF3D7_0732200 (ID#RS100, Rifin), PF3D7_0532900 217

(ID#RS71, Rifin), and PF3D7_0223100 (ID#RS34, Rifin) and accounted for 20.2% of the var- 218

iation. These findings suggest that PfEMP1s domains are major determinants of the vari- 219

ability in the dataset, and are consistent with those of logistic regression model. 220

221

Other areas that need to be improved:

*Authors stated the following: Seroprevalence of the proteins somehow varied between the protein groups; 178

BSP ranged from (0 to 100%), CIDR (22-90%), DBL (0-100%), RIFIN (5-100%), STEVOR (0- 179

95%), and SURFIN (26-100%), however, the difference was not significant.

Are these in Asy or in Sym or collectively?  Be more explicit in the text and in the table. And it is confusing that VSAs are not outlined in the text or table etc as the abstract suggests. Clearer explaination of the VSAs in the abstract, intro, tables are needed as the authors are making a distinction of VSAs and BSPs. This needs to be clearer in the manuscript.

* Where is this data for the following? The breadth of antibody responses (number of antigens recognized by an individual) and geometric means of ASC of all antigens per individual did not correlate with participants age.

Figure 2.

*What are these antigens?

The 268 antigens exhibited a significant negative association with malaria outcomes.

*Why only ten proteins displayed in Figure 2B?

*Authors state the following but where are the readers to look to understand the results and to justify the statement.

ORs of 2 RIFINs 197

(PF3D7_1400400; RS215 and PF3D7_0732400; RS103) and 1 (PF3D7_1400400; RS215) anti- 198

gens were significantly greater than 1, suggesting opposing clinical malaria protection, by 199

both univariate and multivariate logistic regression analysis.

Why did the authors use the following?

In both populations, an antigen was defined as immunoreactive if present in more than 10% of samples assayed [11]

*Authors state the following, but where is this analysis and why not all 691 proteins used?:

Out of 579 antigens measured in both studies, 566 antigens were commonly found to be immunoreactive.

Difficult sentence, please reword and be very explicit here in your description.

The percent seroprevalence derived from Thai and Ugandan samples were strongly positively correlated (Figure 4A).

Same comments as above. Difficult sentence, please reword and be very explicit here in your description.

Again same comment as above: Difficult sentence, please reword and be very explicit here in your description.

The percent seroprevalences against blood stage proteins (BSP) were the most strongly correlated in other group of antigens (Figure 4B).

Same comments as above: Difficult sentence, please reword and be very explicit here in your description.

Unexpectedly, those against PfEMP1 domains were also significantly correlated (Figure 4CD).

Same comments as above: Difficult sentence, please reword and be very explicit here in your description.

Especially, there was a strong correlation between % seroprevalence for the overlapping immunore- 233

active antigens (n = 163) derived from DBL domain between the 2 sample sets (Figure 4B, 234

r = 0.7219, P<0.0001).

Figure 3. Text is too small to read.

There are labels of Figure 1, Figure 2 and Figuire 3 (spelling), Figure 4 in the figures themselves which should not be.

Supplementary material are traditionally not in the many manuscript but in another file.

Discussion.

In general, authors tend to cite their own work a lot, but what about other people’s work? In addition, there is a lot of result description in the discussion that is repeating what the results stated. Really the discussion section as the author’s guidelines state:

Discussion: Authors should discuss the results and how they can be interpreted in perspective of previous studies and of the working hypotheses. The findings and their implications should be discussed in the broadest context possible and limitations of the work highlighted. Future research directions may also be mentioned.

Hard sentence:

We did not include RH2b in this study to measure antibody levels against as many as VSA antigens, however, MSP3, MSP7 and MSPDBL1 were also significantly associated with reduced risk of clinical malaria, consistent with previous study (Table S2, S3).

In conclusion, despite differences in transmission settings, and human and parasite 410

genetic backgrounds, similar antigens may be central in determining infection outcomes, 411

and may be useful as serological markers of symptomatic infections and blood-stage vac- 412

cine candidates.

See the comments to the authors for this. 

Author Response

Manuscript titled ” High-throughput antibody profiling identifies targets of protective immunity against P. falciparum malaria in Thailand” aimed at assessing the antibody response in individuals from a low transmission area for malaria in Thailand.

With the use of their pre-validated library of P. falciparum recombinant proteins (n=691), authors were able to investigate which peptides / proteins were mostly immunogenic by asymptomatic  and symptomatic malaria patients. Authors had great intent but there are major issued that need to be address so that we can value the research presented in the manuscript. Overall the manuscript is often difficult to read as the language is difficult, awkward, vague or details are really missing. Often there are areas that make the manuscript difficult to follow as it is very convoluted in sentence structure and themes. Mostly the results are not descriptive enough to have the impact or clarity that is needed to justify many statements made from the authors.

Thank you very much for your valuable comments. We have carefully copy-edited the whole manuscript to improve clarity.

More details are in the following:

Difficult language or other issues in Abstract: Please reword

*As the consented efforts optimized towards decreasing malaria transmission and for eventual malaria elimination are implemented, driving previously malaria high to low-transmission regions, it is necessary to understand how interventions affect the naturally acquired protective immunity.

We have updated the abstract in the revised manuscript.

*samples collected from residents of a malaria low-transmission region of Thailand

 We have updated it in the revised manuscript.

*We observed that of the variable surface antigens and merozoites stage proteins included in the library,

antibodies to 268 antigens significantly correlated to the absence of symptomatic malaria in univari-

ate analysis.

We have updated in the revised manuscript.

*Why mention the following? “The antibody assays were conducted using the AlphaScreen system” If important enough to be put into the abstract than authors need to describe what this system is.

The description of AlphaScreen system is added into Abstract, introduction and discussion. Please refer to line no 22, 23, 68, 69, 70, 335, 336, 337, 338, 339, 340 and citation no [19].

* The top screened antigens were mainly PfEMP1 domains. What do you mean by screened here as it probable does mean what you want. Do you mean the antigens with the most immune responses against it mainly PfEMO1 domains. Or?

Please define what PfEMO1 domains are.

We have updated it in the revised manuscript to improve clarity.

Too vague statement. Need to add context or explicit meaning “These observations were consistent with our recent findings with samples from a malaria hyper-endemic region, suggesting that similar antigens could be central in determining infection outcomes in diverse populations.”

For instance, what is the “recent findings” exactly?  Which similar antigens?

We have updated it in the revised manuscript.

* It remains important to functionally characterize the unknown proteins as potential correlates of protection or as vaccine targets. What do you mean by unknown proteins

“unknown proteins” were those proteins that do not have a defined function.

https://plasmodb.org/plasmo/app 

We removed the sentence to improve clarity.

Introduction

Clarity is needed with explicit context and better language use:

*Developing an effective malaria vaccine is a crucial objective… Crucial objective for whom?

It is crucial objective for those researchers who are working on the development of effective malaria vaccine, which will eventually decrease mortality and eradicate malaria globally.

We have updated the revised manuscript based on your comment.

*This goal is supported by evidence demonstrating that individuals living in regions where malaria is endemic develop a partial immunity mediated by antibodies [2, 3]. Goal to develop a vaccine or what do the authors mean here as this statement does not make sense-

Yes, the goal is to develop a vaccine. Furthermore, the statement “This goal is supported by evidence demonstrating that individuals living in regions where malaria is endemic develop a partial immunity mediated by antibodies [2, 3]” enlightens the importance of antibodies in developing protective immunity against malaria.

We have updated it in the revised manuscript based on your comment.

*Awkward sentence that does not make sense. Since symptomatic manifestation of malaria is primarily attributed to asexual blood-stage antigens of Plasmodium falciparum, which have the potential to stimulate a strong immune reaction, these discoveries have led to the implementation of various methods for exploring discovery and development of new vaccine candidates.

Thank you. We have updated the revised manuscript.

*Citations are needed in these sentences: Targeted immuno-epidemiological investigations have been instrumental in detecting, recognizing, and characterizing numerous dominant vaccine candidates currently 45

under development. In particular, immunoscreening techniques that permit the simulta- 46

neous assessment of various proteins as prospective vaccine candidates or immune corre- 47

lates have emerged as a crucial strategy.

These were original sentences, so they don’t need citations.

*Grammar in this sentence and more explicit understanding of what these proteins are, is needed, for example spell out the acronyms. Several proteins, such as MSP1, MSP2, MSP3, AMA1 Reviewed in [6], SERA5 [7] and GLURP [8] have been identified through such approaches.

We have updated it in the revised manuscript based on your comment.

*More description is needed for the AlphaScreen platform Line 54.

The description of AlphaScreen system is added into the abstract, introduction and discussion. Please refer to lines no 22, 23, 68, 69, 70, 335, 336, 337, 338, 339, 340, and citation no [19].

*Citation is needed in the following:

Recent investigations have focused on identifying parasite protein families that are localized to the surface 56

of infected red blood cells, i.e., variable surface antigens (VSAs), including all the domains 57

of erythrocyte membrane protein 1 (PfEMP1), repetitive interspersed family (RIFINs) pro- 58

teins, sub telomeric variable open reading frame (STEVORs), and surface-associated in- 59

terspersed gene family (SURFINs), as well as blood stage proteins (BSP) encoded in the 60

3D7 strain genome.

We have included it in the updated manuscript.

*Confused as to whom did the following? Was it the authors or others, orIf others, citation is needed and need to not use short hand  (and in other areas of the paper) here as it is more difficult to read. Reword please.

These proteins were evaluated against serum samples obtained from individuals residing in a malaria-endemic region of Uganda, where an entomological inoculation rate (EIR) of > 100 infective bites/year.

We have added the citation in the updated manuscript.

* This sentence is confusing. What is meant by “three different clinical malaria definitions?”

Of the proteins tested, 22 induced antibody responses significantly linked to a reduced risk of malaria episodes, as determined by three different clinical malaria definitions based on parasite density (1,000/2,500/5,000 parasites/μl blood) along with the presence of fever [11].

This sentence “three different clinical malaria definitions” refers to the criteria used to define a malaria episode based on parasite density (1000 parasites/ul, 2500 parasites/ul and 5000 parasites/ul). These definitions were used to determine whether an individual is experiencing a malaria infection.

* Why is it important to evaluate…. In the sentence below?

It’s important to evaluate the effect of reduced malaria transmission on the P. falciparum-specific antibody responses in the affected populations.

Evaluating antibody responses provides insights into the immune status of individuals in specific populations. Moreover, assessing the impact of reduced malaria transmission on P. falciparum specific antibodies can help to understand better the effectiveness of natural or induced immunity against malaria in lower transmission regions.

*Punctuation is wrong in the sentence. Please correct. And citation [14] should be added to the data in this sentence.

In line with this idea, we have previously profiled P. falciparum BSPs against clinical malaria by analyzing serum samples obtained from Thailand; a region of low malaria endemicity where EIR of 1–3 infective bites/year [13].

We have included it in the revised manuscript.

*Confusing statements.  Why are the authors looking into VSA when they have cited their own work in the sentence and already looking into this?  Please really explain what is the point of looking into VSA if you have already done this as cited by yourselves.

Our previous study, which was cited for variant surface antigens (VSA), utilizes serum samples from Uganda (endemic region of malaria), whereas the current study employs serum samples from Thailand (lower transmission region of malaria). Moreover, we were interested in VSA because they are located on the surface of infected red blood cells (iRBCs) and a major target of protective immunity.

However, the antigens used for the study did not include VSAs, which could be promising seromarkers of reduced risk of malaria episodes proved by our recent studies [11, 15].

This statement means that in our previous study conducted with serum samples from Thailand. We utilize a protein library including only asexual blood stage proteins but in our current study, we enlarge our library to include both asexual blood stage proteins and VSA.

And

we sought to comprehensively assess the antimalarial immune profiles in residents of a low malaria transmission region in Thailand as used previously [14].

This statement means that for profiling antibody responses against malaria same serum samples were used in the current and previous study.

And

The analysis targeted 691 P. falciparum antigens, including VSAs and BSPs, with the AlphaScreen platform.

This statement means that 691 P. falciparum antigens were analyzed using AlphaScreen system.

*What are the 691 antigens exactly?

691 antigens were malarial proteins that belong to the asexual blood stage of the parasite. These include PfEMP1 (Plasmodium falciparum erythrocyte membrane protein 1), BSPs (Blood stage proteins), Rifins (repetitive interspersed family of proteins), Stevors (Sub-telomeric variable open reading frame) and Surfins (Surface associated interspersed family of proteins).

*Why use AlphaScreen platform and not other systems?

We have added the following sentences to answer this question in the discussion.

AlphaScreen has been widely used for the detection of protein-protein, protein-DNA, and protein-RNA interactions, for screening of proteins interacting with the target proteins, inhibitors of the target complex of proteins, and so on. Because AlphaScreen has been used for the detection of protein-protein interactions, including antigen-antibody interactions, in particular the combination of biotinylated protein library produced by WGCFS, we applied the method in our studies.

We have also confirmed significant correlation between AlphaScreen and ELISA titers. Spearman r = 0.8266, P < 0.0001 (Sakamoto et al.,https://doi.org/10.1016/j.parint.2017.12.002)

Methods

Methods sections needs to be written so some can repeat them and reproduce the results demonstrated from the authors. Right now this is not the case. Controls are missing, explicit details are missing in most methods and information on repeats are not provided. Also authors must clarify if proteins were used or peptides in all parts of the manuscript.

We have updated it in the revised manuscript.

More details in the following:

Awkward sentences. Please clarify.

*Do you mean “The absence of fever was used to characterize ……

The absence of fever characterized asymptomatic P. falciparum malaria (< 37.5 ) during the time of sample collection (Asy) [13].

We have updated the manuscript to clarify.

* Although, in our previous study [14], we used 19 serum samples in Asy group, the number

of serum samples with enough amount left is 17 for this study.

This statement means that our previous study includes total 19 serum samples in asymptomatic group but in our current study we only have 17 serum samples in asymptomatic group because the two samples don’t have enough amount for use.

* before the one corresponding to the sample collection.

This statement means that symptomatic cases enrolled in this study had at least one confirmed and recorded P. falciparum malaria episode before serum samples were collected.

* What is the ethical approval number. Please put like others here in the methods section.

The Institutional Review Boards of Ehime University Hospital, Japan, 104

also approved the protocol for using the archived serum samples [14].

All ethical approval numbers are now included in the text

*What is (9) in this statement?  Is it the citation, if so then put it correctly in the text.

The protein library used in this study included a total of 691 proteins, which repre- 108

sented various protein families of P. falciparum, such as blood stage proteins (BSP; n=157) 109

selected from our previous study based on their significant immunoreactivities (9), cyste- 110

ine-rich inter-domain regions of P. falciparum erythrocyte membrane protein 1 111

(PfEMP1)(CIDR; n=108), Duffy binding-like domains of PfEMP1 (DBL; n=163), repetitive 112

interspersed family proteins (RIFIN; n=178), subtelomeric variable open reading frame 113

proteins (STEVOR; n=53) and surface-associated interspersed gene family proteins 114

(SURFIN; n=32).

Reference has been updated as [13].

 *Authors stated

“The analysis targeted 691 P. falciparum antigens, including VSAs and BSPs, with the AlphaS- 81

creen platform.” But in the methods sections, the VSAs are not defined. Why?

Variable surface antigens (VSAs) are defined in the introduction section. Please refer to line no 72, 73, 74, 75, 76 and 77.

*What genes were cloned and how? PCR protocol, Primers used, restriction enzyme used etc…

The target genes were cloned into the pEU plasmid vector (CellFree Sciences, Matsuyama, Japan).

We have included in the revised manuscript and references added. For clarity, the design of these plasmids and initial expression of the recombinant proteins was already published as referenced.

*Italicize in vitro.

 Updated as your suggestion.

* How and which ones because why clone things if you are synthetically producing the proteins?  Or provide a citation.

Mono-biotinylated recombinant proteins were synthesized in vitro using a semi-automated GenDecoder 1000 robotic protein synthesizer (CellFree Sciences).

The proteins were not synthetically produced but rather expressed using the wheat germ cell-free system in a semi-automated process using a machine called “GenDecoder 1000 robotic protein synthesizer”.

*Is there a citation for the AlphaScreen protocol?  If not then why the 1/4000 dilution of sera with proteins?  What concentration of the proteins were used? What are the positive controls for the binding experiment and negative controls for the experiment? What was the buffers used etc. As for the secondary antibody, what was its dilution, its controls that it is working and buffer used.

The citation for AlphaScreen protocol is located in line 156. We have also added the details of the controls used. “The biotinylated IgG was also used as the positive control while translation mixtures with wheat germ extract (WGE) but lacking the parasite mRNA transcript was used as the negative control”. We use AlphaScreen acceptor beads conjugated protein G, thus no secondary antibody was used.

Because sera contain inhibitors for AlphaScreen, we should use 1/10000 dilution of sera for this experiment. The proteins are not purified, but based on previous studies, WGCFS could synthesize (express) plasmodial protein ranging from 0.6 μg/mL to 53 μg/mL (http://iai.asm.org/content/suppl/2008/03/12/76.4.1702.DC1/IAI_CellFree_SuppleTableS1.pdf).

The translation mixture was diluted 250 times, thus final concentration of proteins ranged between 2.4 ng/ml and 212 ng/ml, i.e. 60 pg to 530 pg protein exists in each 25 ul reaction.

*Why was the following used as the cut-off mark as this is not the convention?

The protein seropositivity cut-off value to human sera was defined at the lowest 141

non-negative ASC value from the assayed samples.

This cut-off mark is determined by the ASC value from the lowest value of the wells with lowest concentration of the standards included in the current experiment [20]. Using this cut-off mark for seropositivity, we may have more “positives”. However, the purpose of this study is to compare IgG responses across different malaria outcomes, thus, this is not currently our greatest concern.

*and why did the authors have this evaluation?

A protein was considered immunore-active if more than 10% of the volunteers had ASC levels above the seropositivity cut-off 143

Value.

This (10%) is an arbitrary cut-off point that is more or less accepted for this type of screening work. A reference has been added.

*Where are the healthy control group in this study?

A Healthy non-malaria control group was not part of this study. The ability to control infection outcomes is considered an important step in acquiring naturally acquired immunity.

Results

In general results should be easy to follow by having the figure, table or supplementary data cited in the text and this is not the case in most places in the result section. Authors have just put at the end of the results the figures and tables and this makes it very difficult to follow the authors.

In addition, the legend should be descriptive enough that the figure or table can stand on its own and be understood. Please improve the legends. In addition, Figures and tables must be place near the text when it is first mention. Reader should not have to go down three pages to see the evidence in the result text. Lastly, it is not enough to superficially mention the result as in the following “The top significant antigens selected in the univariate analysis included PfEMP1s (DBL and CIDR) (Figure 2, Table 2, Table S2).

Thank you very much, we have updated the manuscript as your valuable comments. We improved the legends accordingly.

Authors need to describe the results so that we can understand their meaning and significance. Tables and figures should also be descriptive for this too. For example, why do the authors state this that the top….. include PfEMP….? Be exact as figure 2 has two panels and B) only mentions ten proteins where as Table 2 has 15 peptides. Why are there not 15 proteins mentioned in figure 2? Same feedback for all the results.

Thank you for pointing this out. Now Table 2 has 10 antigens. These are now described well in updated in the revised manuscript.

However, the supplementary data remains as recommended by the journal.

But Authors did well in describing the following, and every panel was describes:

The analysis indicated 205

that the first three components (with eigenvalues of 238, 106, and 34, respectively) ac- 206

counted for 54.4% of the total variation in this data (Figure 3A). Dim 1 vs 2 (Figure 3B) 207

explained 49.5 % of the variability and the main contributors were 208

PF3D7_0632500_DBLγ1 (ID#DC109), PF3D7_1240300_CIDRβ1 (ID#DC246), 209

PF3D7_0712900_CIDRβ1 (ID#DC149), PF3D7_0800300_CIDRα1.6 (ID#DC166), 210

PF3D7_0712600_CIDRα3.1 (ID#DC139). Dim 1 vs 3 (Figure 3C) explained 39.1% of data 211

variability, with the strongest loading arising from antibodies against 212

PF3D7_0632500_DBLγ1 (ID#DC109), PF3D7_1240300_CIDRβ1 (ID#DC246), 213

PF3D7_0425800_CIDRα1.6 (ID#DC71), PF3D7_0712600_CIDRα3.1 (ID#DC139), 214

PF3D7_1300300_CIDRβ3 (ID#DC268). Dim 2 vs 3 (Figure 3D) was mainly reflective of 215

PF3D7_0402100 (ID#EH9, Plasmodium exported protein PHISTb, unknown function), 216

PF3D7_1040400 (ID#RS156, Rifin), PF3D7_0732200 (ID#RS100, Rifin), PF3D7_0532900 217

(ID#RS71, Rifin), and PF3D7_0223100 (ID#RS34, Rifin) and accounted for 20.2% of the var- 218

iation. These findings suggest that PfEMP1s domains are major determinants of the vari- 219

ability in the dataset, and are consistent with those of logistic regression model. 220

221

Thank you.

Other areas that need to be improved:

*Authors stated the following: Seroprevalence of the proteins somehow varied between the protein groups; 178

BSP ranged from (0 to 100%), CIDR (22-90%), DBL (0-100%), RIFIN (5-100%), STEVOR (0- 179

95%), and SURFIN (26-100%), however, the difference was not significant.

Are these in Asy or in Sym or collectively?  Be more explicit in the text and in the table. And it is confusing that VSAs are not outlined in the text or table etc as the abstract suggests. Clearer explaination of the VSAs in the abstract, intro, tables are needed as the authors are making a distinction of VSAs and BSPs. This needs to be clearer in the manuscript.

These were collectively across Asy and Sym groups. Now the sentence is "Seroprevalence of the proteins in all assessed participants varied between the protein groups; BSP ranged from (0 100%), CIDR (22-90%), DBL (0-100%), RIFIN (5-100%), STEVOR (0-95%), and SURFIN (26-100%), however, the difference was not significant."

VSAs are described in the introduction section. Please refer to lines no, 72, 73, 74, 75, 76 and 77.

“Recent investigations have focused on identifying parasite protein families that are localized to the surface of infected red blood cells, i.e., variable surface antigens (VSAs), (that include all the domains of erythrocyte membrane protein 1 (PfEMP1), repetitive interspersed family (RIFINs) proteins, sub telomeric variable open reading frame (STEVORs), and surface-associated interspersed gene family (SURFINs)), as well as other blood stage proteins (here referred to as BSP) encoded in the 3D7 strain genome”

* Where is this data for the following? The breadth of antibody responses (number of antigens recognized by an individual) and geometric means of ASC of all antigens per individual did not correlate with participants age.

The data was not shown. We added scattered plots for the results (Figure S2).

Figure 2.

*What are these antigens?

The 268 antigens exhibited a significant negative association with malaria outcomes.

The 268 antigens were  Plasmodium falciparum erythrocyte membrane protein 1 (PfEMP1) and blood stage proteins (BSPs) which were associated with reduced risk of symptomatic malaria. The full list is shown in Table S2.

*Why only ten proteins displayed in Figure 2B?

Only the top 10 proteins are presented here. The full list is shown in Table S2. We adjusted the numbers of antigens in Table 2.

*Authors state the following but where are the readers to look to understand the results and to justify the statement.

ORs of 2 RIFINs 197

(PF3D7_1400400; RS215 and PF3D7_0732400; RS103) and 1 (PF3D7_1400400; RS215) anti- 198

gens were significantly greater than 1, suggesting opposing clinical malaria protection, by 199

both univariate and multivariate logistic regression analysis.

The text has been updated.

Why did the authors use the following?

In both populations, an antigen was defined as immunoreactive if present in more than 10% of samples assayed [11]

This has been changed to read “studies” as described in the preceding statement.

*Authors state the following, but where is this analysis and why not all 691 proteins used?:

Out of 579 antigens measured in both studies, 566 antigens were commonly found to be immunoreactive.

This has been adjusted to read “Of all the antigens measured in both studies (579 of 691), 566 antigens were commonly found to be immunoreactive”.

 Difficult sentence, please reword and be very explicit here in your description.

The percent seroprevalence derived from Thai and Ugandan samples were strongly positively correlated (Figure 4A).

It’s now updated in the revised manuscript.

Same comments as above. Difficult sentence, please reword and be very explicit here in your description.

Again same comment as above: Difficult sentence, please reword and be very explicit here in your description.

The percent seroprevalences against blood stage proteins (BSP) were the most strongly correlated in other group of antigens (Figure 4B).

We have updated it in the revised manuscript.

Same comments as above: Difficult sentence, please reword and be very explicit here in your description.

Unexpectedly, those against PfEMP1 domains were also significantly correlated (Figure 4CD).

 We have updated it in the revised manuscript.

Same comments as above: Difficult sentence, please reword and be very explicit here in your description.

Especially, there was a strong correlation between % seroprevalence for the overlapping immunore- 233

active antigens (n = 163) derived from DBL domain between the 2 sample sets (Figure 4B, 234

r = 0.7219, P<0.0001).

We have updated it in the revised manuscript.

Figure 3. Text is too small to read.

We have updated it in the revised manuscript.

There are labels of Figure 1, Figure 2 and Figuire 3 (spelling), Figure 4 in the figures themselves which should not be.

Supplementary material are traditionally not in the many manuscript but in another file.

We have updated it in the revised manuscript.

Discussion.

In general, authors tend to cite their own work a lot, but what about other people’s work? In addition, there is a lot of result description in the discussion that is repeating what the results stated. Really the discussion section as the author’s guidelines state:

Discussion: Authors should discuss the results and how they can be interpreted in perspective of previous studies and of the working hypotheses. The findings and their implications should be discussed in the broadest context possible and limitations of the work highlighted. Future research directions may also be mentioned.

In the discussion section, we discussed the results and their connection with previous findings have also been explained. We also write about the limitations of our work, whereas future directions are mentioned in the conclusion section.

Hard sentence:

We did not include RH2b in this study to measure antibody levels against as many as VSA antigens, however, MSP3, MSP7 and MSPDBL1 were also significantly associated with reduced risk of clinical malaria, consistent with previous study (Table S2, S3).

It’s now updated in the revised manuscript.

In conclusion, despite differences in transmission settings, and human and parasite 410

genetic backgrounds, similar antigens may be central in determining infection outcomes, 411

and may be useful as serological markers of symptomatic infections and blood-stage vac- 412

cine candidates.

We have updated it in the revised manuscript.

Round 2

Reviewer 2 Report

I appreciate the efforts made by the authors to enhance the manuscript. They have thoroughly addressed all of my comments, and as a result, I no longer have any objections to the publication of the article.

Author Response

Thank you very much for your kind review to improve this manuscript.

Reviewer 3 Report

At this stage of the review it is imperative that the author’s follow the instructions of the journal for the manuscript and correct all points provided by the reviewers. It is also imperative that there is a cleaned version of the manuscript also uploaded so that the review can easily see the overall manuscript without the track changes. The authors have indeed started to correct all the issues outlined in the first review, but there are still major issues not addressed by the authors; see below.

With this second version there are still many untidy items in the manuscript that draws away substantially from the credibility of the science. A thorough combing through to fix all the mistakes are need at this stage. It would be a pity if this manuscript is finally rejected due to many mistakes. For example, all bacterial names are to be italicized and acronyms are to be spelled out then used. Figure and table legends are to make sure the figures and tables are standing out on their own and the readers understands them and in the same table there are different fonts.

Abstract:

Language issues with the following:

Line 25-27 starting with “We conducted” the antibody assays were conducted using the AlphaScreen system. We ob-25 served; a homogeneous proximity-based bead assay, a high-throughput system that detects protein 26 interactions.

Line 27-29 starting with “Our findings revealed”. Reword the following  “in univariate analysis” to in an univariate analysis

*Acronyms are spelled out initially and in each section to make the understanding of the content more clear. For example, PfEMP1

Introduction

Difficult language:

This has informed the need

Remove the word Reviewed in line 74.

Punctuation in the following is wrong. Please correct.

In addition, AlphaScreen system; a homogeneous proximity-based bead assay that de-83 tects protein interactions, has been integrated with WGCFS and customed to offer a high-84 throughput antibody assays.

Too many language issues in this sentence

Recent investigations have focused on identifying parasite protein families that are 87 localized to the surface of infected red blood cells, i.e., variable surface antigens (VSAs), (that include all the domains of P. falciparum erythrocyte membrane protein 1 (PfEMP1), repetitive interspersed family (RIFINs) proteins, sub telomeric variable open 90 reading frame (STEVORs), and surface-associated interspersed gene family (SURFINs),)), 91 as well as other blood stage proteins (here referred to as BSP) encoded in the 3D7 strain 92 genome [13].

Language issue

*Im-100 portantly, the 20 selected proteins among 22 were belonged to VSAs. Remove were.

Methods

These individuals 123 were enrolled in an active follow-up study.  What is meant by active follow-up study? 

An asymp-124 tomatic P. falciparum malaria (Asy) case was defined as having a parasitic infection but 125 with no fever (< 37.5 ) duringat the time of sample collection [15]. How do the authors know if there was a parasitic infection in the cases?

Seventeen (17) of the 19 serum samples [16]in Asy group, 127 the number of serum samples, with enough amount left is 17 for this studyserum volume, 128 were used in the current assays.  What do the authors mean here. Was there there 19 cases so samples  and only 17 were used as there was enough serum volume?  Is this correct, then the authors need to make this clearer here as interchanging cases with samples is vague and confusing. Also why was there not enough serum volumn?

* 21 adults diagnosed with uncomplicated symptomatic P. 132 falciparum malaria (Sym) between May and June 2005. Again more information is needed here. What is meant by diagnosed, how was the adults diagnosed, but what criteria or test or?? And authors state uncomplicated symptomatic, but what is really meant by uncomplicated and symptomatic???? Based on what definitions or criteria or citation or guideline, etc?

*Similarly what is meant by “confirmed and recorded P. falciparum malaria episode” How confirmed, what is meant by recorded, etc.

*How is parasitemia elucidated by microscopy? Visually with IFA or what?

* The Institutional Review 141 Boards of Ehime University Hospital, Japan, also approved the protocol for using the ar-142 chived serum samples (Aidaiibyourin 1507005 dated 1 August 2015) [16].  More context is needed for these archived samples. Add it here. For example why was Japanese samples used and how many etc. Where is the data in the results and in discussion?

*What patient consent received for the “Ethics Committee 129 of the Thai Ministry of Public Health and the Institutional Review Board of the Walter 130 Reed Army Institute of Research (WRAIR 778; dated 14 April 2000) “samples?

Results

*Wrong use of () in (30 (range; 11-50) years). Please correct.

*Table 1: what is meant by Median parasitemia % (range)

* with a seroprevalence higher than the defined threshold of 10%. What is meant by defined threshold of 10%?

Figure 1 legend is not with its figure and it needs a Title.  Where in the methods is the cutoff described? For example, how is it made?

Figure 2S is before figure S1. This is incorrect, please correct.

*Author’s stated “all antigens per individual did 227 not correlate with participants age.” How can you say not correlated here?  Please be explicit.

Figure 2 needs a title.

*In figure 2 legend, authors stated “The antibody levels between the 273 Asymptomatic (Asy) and Symptomatic (Sym) groups were compared.” But only ten proteins are listed here. Make the legend clearer please.

* The top 10 significant antigens 249 selected: Please list the ten significant antigens

* these proteins may be acting different compared to the proteins.  What is meant by acting different compared to the proteins (what proteins)?

*Why are some Figure in the text bolded? Please correct.

Discussion

*the in15adence.  What is this in line 451?

*Line 454 why bolded text?

Conclusion

Awkward wording, Put together. Please correct.

*What is meant by these antigens in line 558, from the study or?

*Figure S1 is missing title and legend

*Figure S2. Spell out the acronyms.

*Authors state the following statements but they are contradicting themselves and what is meant by scripts? “All the data used in this study and scripts are available as supplemen-606 tary data. 607

The data is available from the authors on request.

*Punctuation language of the sub-headings is incorrect. Please correct as per author’s instructions.

*Reference are not in the correct format. Please read the author’s instruction for the correct format.

Still needs some work. See comments for authors for details. 

Author Response

At this stage of the review it is imperative that the author’s follow the instructions of the journal for the manuscript and correct all points provided by the reviewers. It is also imperative that there is a cleaned version of the manuscript also uploaded so that the review can easily see the overall manuscript without the track changes. The authors have indeed started to correct all the issues outlined in the first review, but there are still major issues not addressed by the authors; see below.

Thank you very much for your kind review to improve this manuscript.

With this second version there are still many untidy items in the manuscript that draws away substantially from the credibility of the science. A thorough combing through to fix all the mistakes are need at this stage. It would be a pity if this manuscript is finally rejected due to many mistakes. For example, all bacterial names are to be italicized and acronyms are to be spelled out then used. Figure and table legends are to make sure the figures and tables are standing out on their own and the readers understands them and in the same table there are different fonts.

Abstract:

Language issues with the following:

Line 25-27 starting with “We conducted” the antibody assays were conducted using the AlphaScreen system. We ob-25 served; a homogeneous proximity-based bead assay, a high-throughput system that detects protein 26 interactions.

This has been corrected to read:

We conducted the antibody assays using the AlphaScreen system, a high-throughput homogeneous proximity-based bead assay that detects protein interactions.

Line 27-29 starting with “Our findings revealed”. Reword the following  “in univariate analysis” to in an univariate analysis

This has also been corrected to read:

We observed that out of the 691 variable surface and merozoite stage proteins included in the library, antibodies to 268 antigens significantly correlated with the absence of symptomatic malaria in an univariate analysis

*Acronyms are spelled out initially and in each section to make the understanding of the content more clear. For example, PfEMP1

We have updated in the revised manuscript.

Introduction

Difficult language:

This has informed the need

Remove the word Reviewed in line 74.

Done

Punctuation in the following is wrong. Please correct.

“In addition, AlphaScreen system; a homogeneous proximity-based bead assay that de-83 tects protein interactions, has been integrated with WGCFS and customed to offer a high-84 throughput antibody assays.”

Done. Now reads

In addition, AlphaScreen system, a homogeneous proximity-based bead assay that detects protein interactions, has been integrated to WGCFS and customed to offer high-throughput antibody assays.

Too many language issues in this sentence

Recent investigations have focused on identifying parasite protein families that are 87 localized to the surface of infected red blood cells, i.e., variable surface antigens (VSAs), (that include all the domains of P. falciparum erythrocyte membrane protein 1 (PfEMP1), repetitive interspersed family (RIFINs) proteins, sub telomeric variable open 90 reading frame (STEVORs), and surface-associated interspersed gene family (SURFINs),)), 91 as well as other blood stage proteins (here referred to as BSP) encoded in the 3D7 strain 92 genome [13].

Thank you very much for your comments. The sentence has been updated as follows.

Leveraging the availability of a well annotated 3D7 P. falciparum genome, recent investigations have focused on identifying parasite protein that are key targets of antibodies-mediated blood-stage immunity. This includes variable surface antigen (VSA) family proteins localized to the surface of infected red blood cells, such as P. falciparum erythrocyte membrane protein 1 (PfEMP1), repetitive interspersed family (RIFIN) proteins, sub-telomeric variable open reading frame (STEVOR) proteins, and surface-associated interspersed gene family (SURFIN) proteins, as well as other asexual-stage parasite proteins (here referred to as BSP) [13]

Language issue

*Im-100 portantly, the 20 selected proteins among 22 were belonged to VSAs. Remove were.

Done

Methods

These individuals 123 were enrolled in an active follow-up study.  What is meant by active follow-up study? 

An active follow-up study is where study participants are actively contacted as compared to a passive follow-up which is based on the reporting of cases by physicians or hospitals as participants freely come to the clinic when unwell. Active follow-up is important for correcting participants data at pre-defined time points.

An asymptomatic P. falciparum malaria (Asy) case was defined as having a parasitic infection but with no fever (< 37.5 ℃) during at the time of sample collection [15]. How do the authors know if there was a parasitic infection in the cases?

We have added the words “microscopically detectable” for clarity.

An asymptomatic P. falciparum malaria (Asy) case was defined as having a detectable parasitic infection, following a microscopic examination of Giemsa-stained blood films, but with no fever (< 37.5 ) at the time of sample collection [15]

Seventeen (17) of the 19 serum samples [16] in Asy group, 127 the number of serum samples, with enough amount left is 17 for this study serum volume, 128 were used in the current assays.  What do the authors mean here. Was there there 19 cases so samples  and only 17 were used as there was enough serum volume?  Is this correct, then the authors need to make this clearer here as interchanging cases with samples is vague and confusing. Also why was there not enough serum volumn?

Yes. Two (2) of the 19 serum samples were almost completely used during our previous study. We have revised the sentence to read:

Although samples from 19 participants were collected, only 17 were available for the current assays due to serum volume limitation.

* 21 adults diagnosed with uncomplicated symptomatic P. 132 falciparum malaria (Sym) between May and June 2005. Again more information is needed here. What is meant by diagnosed, how was the adults diagnosed, but what criteria or test or??

Microscopy was the standard for diagnosis. All the definitions used in the text are provided and are based on WHO standard definitions.

The parasites on a blood smear were observed by microscopy, followed by Giemsa staining (https://apps.who.int/iris/bitstream/handle/10665/162441/9789241549127_eng.pdf).

Sym malaria was defined as the presence of fever (> 37.5 °C) during sample collection and any parasitemia detectable by microscopy with no evidence or sign of severe disease as defined by WHO

And authors state uncomplicated symptomatic, but what is really meant by uncomplicated and symptomatic???? Based on what definitions or criteria or citation or guideline, etc?

Again, all the definitions used in the text are provided and are based on WHO standard definitions.

Uncomplicated symptomatic malaria is malaria with no signs of severity and/or evidence of vital organ dysfunction. (https://apps.who.int/iris/bitstream/handle/10665/162441/9789241549127_eng.pdf)

*Similarly what is meant by “confirmed and recorded P. falciparum malaria episode” How confirmed, what is meant by recorded, etc.

This has been corrected to read as below. The word microscopy had been left out reduce redundancy, since are these are standard definitions in malaria research. Now reads

All 21 Sym cases had at least one microscopically confirmed P. falciparum malaria episode in a clinical record before the one corresponding to the sample collection. Sym malaria was defined as the presence of fever (> 37.5 °C) during sample collection and any parasitemia detectable by microscopy.

*How is parasitemia elucidated by microscopy? Visually with IFA or what?

Revised to read as:

An asymptomatic P. falciparum malaria (Asy) case was defined as having a detectable parasitic infection, following a microscopic examination of Giemsa-stained blood films or nested PCR positive, but with no fever (< 37.5 ) at the time of sample collection [15].

The parasites on a blood smear were observed by microscopy, followed by Giemsa staining. (https://apps.who.int/iris/bitstream/handle/10665/162441/9789241549127_eng.pdf).

* The Institutional Review 141 Boards of Ehime University Hospital, Japan, also approved the protocol for using the ar-142 chived serum samples (Aidaiibyourin 1507005 dated 1 August 2015) [16].  More context is needed for these archived samples. Add it here. For example why was Japanese samples used and how many etc. Where is the data in the results and in discussion?

No Japanese samples were used in this study. The basic ethics standards demand that ethics approval is obtained from the institution where the samples are to be assayed. That was done. The Ehime University Hospital IRB approval is for the use of the Thai samples in Japan.

*What patient consent received for the “Ethics Committee 129 of the Thai Ministry of Public Health and the Institutional Review Board of the Walter 130 Reed Army Institute of Research (WRAIR 778; dated 14 April 2000) “samples?

This approval was for recruiting and collecting samples from residents of Kong Mong Tha, located in Kanchanaburi Province in western Thailand.

For clarity, “Consent” means that each participant voluntarily and legally agreed to participate in the study as required by the principles of the Declaration of Helsinki.

Results

*Wrong use of () in (30 (range; 11-50) years). Please correct.

Revised to read as:

Serum samples were obtained from adult participants in Thailand, as shown in Table 1. The median age of the Asy group, 30 (range: 11-50) years was higher than the Sym group 23 (range: 17-50) years.

*Table 1: what is meant by Median parasitemia % (range)

The parasitemia data is shown by percentage, and the ranges are shown in parentheses.

* with a seroprevalence higher than the defined threshold of 10%. What is meant by defined threshold of 10%?

The defined cutoff value of 10%

Figure 1 legend is not with its figure and it needs a Title. 

Thank you. Now it is included in figure and added a title.

Figure 1. Immunoreactivity of the P. falciparum antigens against Thai serum samples. A) The median AlphaScreen counts (ASC), indicating antibody responses against P. falciparum antigens were compared between two sample groups: Asymptomatic (Asy, n = 17) and Symptomatic (Sym, n = 21) groups. The median reactivity of individuals is represented by dots, while the median of each group is depicted by black bars. (B) The seroprevalence of the specified P. falciparum proteins was determined using Thai serum samples (n =38). The median seroprevalence of the group is represented by black bars. A dashed red horizontal line at 10% seroprevalence serves as the cutoff value, indicating the threshold for defining an immunoreactive antigen used for subsequent statistical analysis. Out of the 691 proteins tested, 674 exhibited seroprevalence above 10% seroprevalence.

Where in the methods is the cutoff described? For example, how is it made?

The cutoff value was determined by the lowest signals derived from the standards. The description is provided in the methods section.

A protein was considered immunoreactive if more than 10% of the volunteers had ASC levels above the seropositivity cutoff value [20].

Figure 2S is before Figure S1. This is incorrect, please correct.

Thank you. Now it has been updated as per your suggestion.

*Author’s stated “all antigens per individual did 227 not correlate with participants age.” How can you say not correlated here?  Please be explicit.

This has been updated to read as below.

We observed a significant correlation between participants age with neither the breadth of antibody responses (number of antigens recognized by an individual, Spearman rank correlation tests; r=0.0157, p=0.92) nor the geometric means of ASC of all antigens recognized by an individual (r=0.1811, p=0.27).

Figure 2 needs a title.

*In figure 2 legend, authors stated “The antibody levels between the 273 Asymptomatic (Asy) and Symptomatic (Sym) groups were compared.” But only ten proteins are listed here. Make the legend clearer please. * The top 10 significant antigens 249 selected: Please list the ten significant antigens

Updated as shown below

Figure 2: Comparison of the antibody responses with the risk of clinical malaria A) The association between antibody responses and the risk of clinical malaria as analyzed by univariate logistic regression. Black dots indicate the odd ratios, and error bars indicate a 95% confidence interval, and the red vertical dashed line represents an odds ratio of 1. The 268 antigens exhibited a significant negative association with malaria outcomes shown in the upper section. The very high upper confidence intervals derived from 37 antigens at the bottom are truncated in the figure. The complete list of all proteins is included in Table S2. B) The antibody levels between the Asymptomatic (Asy) and Symptomatic (Sym) groups were compared. The log 10 transformed AlphaScreen Count (ASC) for the selected 10 antigens are represented by dots and box plots, namely PF3D7_1219300_CIDRα3.4; DC239, PF3D7_1150400_DBLβ3; DC217, PF3D7_1240300_CIDRα4; DC243, PF3D7_0412900_CIDRα3.1; DC47, PF3D7_0223500_CIDRα3.4; DC15, PF3D7_0421300_DBLα0.24; DC66, PF3D7_0937800_DBLδ1; DC195, PF3D7_1035500 (MSP6); WE18, PF3D7_1400400; RS215 and PF3D7_0732400; RS103. Blue and red indicate Asy and Sym cases, respectively. The antigens names consist of PlasmoDB gene ID plus the expressed domain or protein name.

* these proteins may be acting different compared to the proteins.  What is meant by acting different compared to the proteins (what proteins)?

Thank you. The sentence has been updated as follows.

ORs of 2 RIFINs (PF3D7_1400400; RS215 and PF3D7_0732400; RS103) were significantly greater than 1 (Table S2 Table S3), indicating these proteins may be acting different compared to the other proteins.

*Why are some Figure in the text bolded? Please correct.

This is now corrected as per your suggestion.

Discussion

*the in15adence.  What is this in line 451?

Thank you very much. Now it is updated as follows.

Due to the collective global efforts to alleviate the impact of malaria, the prevalence of P. falciparum infections have considerably decreased in certain African countries and Southeast Asia [14].

*Line 454 why bolded text?

Corrected.

Conclusion

Awkward wording, Put together. Please correct.

Thank you very much. “put together” is removed.

*What is meant by these antigens in line 558, from the study or?

It is updated as not bolded per your suggestion.

Additionally, these target antigens identified by this study

*Figure S1 is missing title and legend

Done

*Figure S2. Spell out the acronyms.

Done

*Authors state the following statements but they are contradicting themselves and what is meant by scripts? “All the data used in this study and scripts are available as supplemen-606 tary data. 607

The data is available from the authors on request.”

Thank you very much for pointing out this error.

All the data used in this study and R-scripts used for the statistical analysis are available from the authors on request.

*Punctuation language of the sub-headings is incorrect. Please correct as per author’s instructions.

Thank you very much. This is now corrected.

3.1. Characteristics of the malaria participants

3.2. Profiling of the human IgG responses to P. falciparum proteins

3.3. Association between antibody responses and risk of clinical malaria

3.4. Principal component analysis of antibody responses

3.5. Comparison of seropositivity between Thai and Ugandan samples

5. Conclusion.

*Reference are not in the correct format. Please read the author’s instruction for the correct format.

We have updated references according to author’s instructions in revised manuscript.